# Machine-Learning-Based Prediction of 1-Year Arrhythmia Recurrence after Ventricular Tachycardia Ablation in Patients with Structural Heart Disease

**DOI:** 10.3390/bioengineering10121386

**Published:** 2023-12-01

**Authors:** Ferenc Komlósi, Patrik Tóth, Gyula Bohus, Péter Vámosi, Márton Tokodi, Nándor Szegedi, Zoltán Salló, Katalin Piros, Péter Perge, István Osztheimer, Pál Ábrahám, Gábor Széplaki, Béla Merkely, László Gellér, Klaudia Vivien Nagy

**Affiliations:** 1Heart and Vascular Center, Semmelweis University, Városmajor u. 68, 1122 Budapest, Hungary; fkomlosi1@gmail.com (F.K.); gyuszkob@gmail.com (G.B.); tokmarton@gmail.com (M.T.); nandorszegedi@gmail.com (N.S.); sallo.zoltan0121@gmail.com (Z.S.); piroskati90@gmail.com (K.P.); peter.perge@gmail.com (P.P.); palabrah@gmail.com (P.Á.); merkely.bela@gmail.com (B.M.); laszlo.geller@gmail.com (L.G.); 2Mater Private Hospital, 69 Eccles St., D07 WKW8 Dublin, Ireland; szeplaki.gabor@gmail.com

**Keywords:** ventricular tachycardia, catheter ablation, recurrence, machine learning, random forest

## Abstract

Background: Ventricular tachycardia (VT) recurrence after catheter ablation remains a concern, emphasizing the need for precise risk assessment. We aimed to use machine learning (ML) to predict 1-month and 1-year VT recurrence following VT ablation. Methods: For 337 patients undergoing VT ablation, we collected 31 parameters including medical history, echocardiography, and procedural data. 17 relevant features were included in the ML-based feature selection, which yielded six and five optimal features for 1-month and 1-year recurrence, respectively. We trained several supervised machine learning models using 10-fold cross-validation for each endpoint. Results: We observed 1-month VT recurrence was observed in 60 (18%) cases and accurately predicted using our model with an area under the receiver operating curve (AUC) of 0.73. Input features used were hemodynamic instability, incessant VT, ICD shock, left ventricular ejection fraction, TAPSE, and non-inducibility of the clinical VT at the end of the procedure. A separate model was trained for 1-year VT recurrence (observed in 117 (35%) cases) with a mean AUC of 0.71. Selected features were hemodynamic instability, the number of inducible VT morphologies, left ventricular systolic diameter, mitral regurgitation, and ICD shock. For both endpoints, a random forest model displayed the highest performance. Conclusions: Our ML models effectively predict VT recurrence post-ablation, aiding in identifying high-risk patients and tailoring follow-up strategies.

## 1. Introduction

Ventricular tachycardia (VT) is a condition associated with poor, possibly lethal clinical outcomes in patients with structural heart disease [1]. Implantable cardioverter defibrillator (ICD) has become the cornerstone of preventing sudden cardiac death (SCD) [2]. However, ICD shocks are associated with increased mortality [3], impaired quality of life, and a negative psychological impact [4]. Therefore, there is an increasing demand for more definitive therapeutic approaches to VTs. Antiarrhythmic drugs (AAD) such as amiodarone are widely accepted as first-line pharmacological therapy and effectively reduce the incidence of malignant ventricular arrhythmias [5]. However, due to a high VT recurrence rate and relatively common side effects, radiofrequency catheter ablation has been proven superior to pharmacological therapy [6,7,8]. During 30 years of practice, numerous developments and modifications have contributed to the increase in success rates of catheter ablation in VTs. In addition to intracardiac electrograms processed in real-time by electroanatomical mapping systems, preoperative electrocardiograms, ICD recordings, and cardiac magnetic resonance imaging (MRI) also play a role in identifying the arrhythmia substrate. Thanks to technological advances, VT ablation is considered a safe and effective therapy for ventricular arrhythmias [9], even in complex scenarios [10].

Despite the improving success rates, arrhythmia recurrence remains an issue, and is associated with higher mortality [11]. Consequently, patients with a high risk of recurrent VTs require a closer follow-up than others, which underlines the necessity for a potent risk stratification system.

Multiple factors have been identified that contribute to the success of the procedure including lower LVEF, multiple inducible VT morphologies, impaired NYHA functional status, presentation with an electrical storm, and non-ischemic VT [11,12,13,14]. Despite the extensive literature on the predictors of VT recurrence, there is a paucity of machine-learning-based analyses. One risk prediction system (the I-VT score) has been proposed by Vergara et al. which uses the clinical data of 1251 patients to create risk stratification models for post-procedural VT recurrence and mortality [15]. Here, the authors use a decision-tree-based classification approach; one which is easy to interpret but carries a higher risk of overfitting compared to more complex machine learning methods. We hypothesize that novel machine learning algorithms can possibly provide a more accurate approach to the question, arming clinicians with a more versatile tool in decision making.

Our complex, single-center study aimed to assess the clinical and procedural factors predicting VT recurrence at two different timepoints during 1-year follow-up using a range of machine learning algorithms. We sought to develop the models with the highest predictive power into a risk stratification system.

## 2. Materials and Methods

### 2.1. Patient Population

In our observational, single-center clinical study, 337 patients were enrolled who underwent their first VT ablation between April 2005 and July 2022. The indication of ablation was based on the current ESC guidelines [16]. The inclusion criteria were detected and clinically proven VT episodes by either a 12-lead surface ECG or ICD recording in patients with structural heart disease, which required radiofrequency (RF) catheter ablation. Structural heart disease included coronary artery disease (CAD) and nonischemic dilated cardiomyopathy (DCM). Patients with idiopathic VT were excluded. All patients who underwent catheter ablation gave written informed consent. This study complied with the Declaration of Helsinki and was approved by the Semmelweis University Research Ethics Committee (Approval Code: 64/2017, Approval Date: 6 April 2017).

### 2.2. Ablation Procedure

Ablation was performed according to the guidelines and recommendations available at the time of the given procedure [16,17].

Right femoral venous and arterial accesses were used for catheter insertion. A quadripolar diagnostic catheter (BIOTRONIK SE & Co. KG, Berlin, Germany) was inserted into the right ventricular apex and a decapolar electrode (Bard, Boston Scientific Marlborough, Massachusetts, USA or St. Jude Scientific, Saint Paul, MN, USA) was inserted into the coronary sinus. The ablation catheter [Therapy Cool Path (St. Jude Scientific), Blazer Open-Irrigated (Boston Scientific, Marlborough, MA, USA), AlCath Black, AlCath Flux Blue (Biotronik, Berlin, Germany), NaviStar ThermoCool, SmartTouch (Biosense Webster, Irvine, CA, USA), TactiCath (Abbott, Chicago, IL, USA)] was introduced via transvenous (right ventricle), transseptal, or retrograde (transaortic) approach. In 25 patients (9.2%), epicardial ablation was performed using subxiphoid access. In all cases, 3D electroanatomical mapping system was used CARTO (Biosense Webster, Irvine, CA, USA) or EnSite (Abbott, Chicago, IL, USA). Programmed extrastimulation up to three extrastimuli was used for induction. During hemodynamically stable VTs, simultaneous activation and substrate mapping were performed to identify exit/isthmus sites; entrainment mapping was also used when possible. Hemodynamically unstable VTs were terminated with electrical cardioversion, followed by substrate mapping in sinus or paced rhythm. Myocardial scar was defined as a bipolar potential amplitude lower than 1.5 mV and dense scar areas were defined as lower than 0.5 mV with no local capture. The main ablation strategy was extensive substrate modification targeting local abnormal ventricular activities (LAVAs) and late potentials (LPs). Additionally, exit and isthmus sites were targeted in hemodynamically tolerated VTs. Acute procedural success was defined as VT non-inducibility after the procedure with programmed extrastimulation up to three beats. In cases where the VT was not inducible during the procedure, procedural success was defined as complete elimination of all LAVAs and LPs.

All patients who did not previously undergo ICD implantation received a secondary prevention ICD after the ablation during index hospitalization.

### 2.3. Collected Data

Most of the data was prospectively collected, while missing data was added based on electronic health records and medical charts in a retrospective fashion; a data availability of over 70% was assured for every feature.

Our database included a total of 31 features, encompassing medical history, echocardiography findings, and procedure-related data. Mortality status was obtained from the National Health Insurance Database of Hungary. VT recurrence was defined as either an ICD recording of a sustained VT episode requiring therapy or an episode of sustained VT recorded on 12-lead ECG. The endpoints were VT recurrence at 1 month and 12 months from the index ablation.

Baseline demographic features included age and sex. Comorbidities such as atrial fibrillation, hypertension, diabetes, chronic pulmonary obstructive disease (COPD), and heart failure (HF) were inferred from past medical history. Coronary artery disease (CAD) was defined as either established chronic coronary syndrome [18], obstructive coronary disease documented by imaging, or the evidence of past coronary revascularization. Pre-ablation ICD and CRT implantation was recorded from the available medical history. In cases of heart failure, the NYHA stage was assessed upon admission for the procedure. Clinical presentation with hemodynamic (HD) instability, ICD shock, incessant VT, and electrical storm were recorded at admission. HD instability was defined as symptomatic hypotension or sudden cardiac death as a result of the clinical VT. ICD shock was defined as at least one appropriate ICD shock preceding the procedure. Incessant VT and VT storm were defined according to contemporary guidelines of the European Society of Cardiology [16,19]. The echocardiographic examinations were performed based on current ASE guidelines within 30 days preceding the procedure [20]. Parameters analyzed in the present study included left ventricular ejection fraction (LVEF) measured with the Simpson method, left ventricular end systolic diameter (LVESD), tricuspid annular plane systolic excursion (TAPSE), diastolic E-wave deceleration time (DT), mitral regurgitation (MR), and the presence of severe tricuspid regurgitation (TR grade III-IV). Previous treatment with amiodarone and/or beta blockers was recorded.

Collected procedural parameters were (a) the number of significantly different QRS morphologies during VT as seen on the surface ECG during induction protocols, (b) clinical VT cycle length (defined as that of the dominant VT morphology during the procedure; where no VT was inducible, the cycle length previously registered by intracardiac electrograms was considered), (c) elimination of the clinical VT (defined as non-inducibility of the dominant clinical VT morphology after the procedure), (d) elimination of all VTs, which was considered when the non-inducibility endpoint was reached during the procedure. Major complications included pericardial effusion causing hemodynamical compromise, intraoperative death, vascular complication requiring vascular surgery, and stroke.

### 2.4. Statistical Study

The statistical analysis was performed using Python 3.10.0 with the SciPy 1.8.0 [21] and lifelines 0.27.0 [22] libraries. The entire patient population was analyzed with respect to the two endpoints, i.e., 1-month and 1-year VT recurrence. First, we divided the study population into two groups, namely patients with and without 1-month VT recurrence. We compared all recorded parameters across the two groups and noted the statistically significant differences. Then we performed the same comparison between patients with and without 1-year VT recurrence. We presented categorical variables as event numbers and percentages. The continuous variables were expressed as medians with interquartile ranges. The variables were reported to deviate from the normal distribution by Kolmogorov-Smirnov’s test. For group comparison, we used the Mann–Whitney test for continuous variables and the Chi-squared test or Fisher’s exact test for dichotomous variables. Results with a *p*-value < 0.05 were stated as statistically significant.

### 2.5. Machine Learning Pipeline

#### 2.5.1. Software and Hardware

The entire analysis was performed in Python version 3.9.18 [23].

The K-Nearest Neighbors (KNN) classifier used for imputation, the standard scaler and min-max scaler used for pre-processing as well as the Random forest (RF), Extreme gradient boosting (XGB), and multilayer perceptron (MLP) classifiers used during the feature selection and model selection process utilize the Scikitlearn 1.1.3 library [24]. During the selection of the final models, a Bayesian grid search from the scikit-optimize 0.9.0 library was used https://zenodo.org/records/1207017 (accessed on 15 June 2023). Oversampling methods such as Synthetic Minority Oversampling Technique (SMOTE), Adaptive Synthetic (ADASYN), and random oversampling (ROS) utilize the imbalanced-learn 0.11.0 library [25]. For SHAP (SHapley Additive exPlanations) analysis, we used the SHAP 0.43.0 package [26].

The computational platform utilized in this study featured an 11th Gen Intel(R) Core(TM) i7-1185G7 processor operating at a base frequency of 3.00 GHz. The system was equipped with 16GB of RAM and ran on a 64-bit operating system with an x64-based processor architecture.

#### 2.5.2. Establishment of Input Features

Before employing machine-learning-based feature selection, it was necessary to decrease the input features due to a disproportion between the recorded features and the number of cases. From the 31 features, 17 were manually selected based on the previous literature and clinical relevance. The main aspects during manual selection were (a) a previously proven statistical relationship between feature and procedural success, (b) significant difference observed during our statistical study, (c) omission of homologous variables.

#### 2.5.3. Data Pre-Processing

Imputation was performed using the K-nearest neighbors (KNN) method. The KNN classifier was configured with a parameter setting of n_neighbors = 10 and employed distance-weighted imputation. We used scaling to avoid any distortion in data processing arising from the different scales of scalar features. The most commonly used scaling methods [27] were evaluated in the pipeline to find the optimal approach. Standard scaling standardizes features by removing the mean and scaling to unit variance, while min-max scaling transforms the features by scaling each feature to a range between 0 and 1. During the selection of the final classifier, we tested each model type with either standard scaling, min-max scaling, or no scaling used. To compensate for the imbalance in the dataset, i.e., the low proportion of cases reaching an endpoint, we tested multiple oversampling methods [28]. ROS (random oversampling) solves the imbalance by randomly re-using cases of the minority class during training, while both SMOTE (Synthetic Minority Oversampling Technique) and ADASYN (Adaptive Synthetic) generate additional cases of the minority group based on existing cases [28].

#### 2.5.4. Feature Selection

Machine-learning-based feature selection was aimed to (a) find the appropriate number of input features for the prediction models and (b) define the input features leading to the most accurate predictions. A flowchart of the feature selection process is presented in Figure 1.

Automated feature selection was performed separately for the 1-month and 1-year recurrence endpoints. Since various model types are considered for the final classifier, we aimed to implement multiple approaches to feature ranking as well. Out of four different approaches, three used the permutation importance method [29], while one used the built-in recursive feature elimination (RFE) function of the XGB model. An additional ranking was created by averaging ranks across the four methods. Each ranking method was applied 100 times, each time using a different random seed, and results were averaged to produce the final ranking orders.

The feature selection approaches used are described in more detail below.

##### Feature Ranking: MLP—Permutation Importance

An optimal MLP model was trained on all cases using all input features using 5-fold cross-validation and the area under the receiver operating curve (AUC) was determined. Then, the analyzed feature was shuffled, and the AUC was re-assessed. The importance of the analyzed feature was inferred from the difference in AUC between the two measurements. This was repeated 100 times and results were averaged.

##### Feature Ranking: RF—Permutation Importance

An optimal RF model was trained on all cases using all input features using 5-fold cross-validation and the area under the receiver operating curve (AUC) was determined. Then, the analyzed feature was shuffled, and the AUC was re-assessed. The importance of the analyzed feature was inferred from the difference in AUC between the two measurements. This was repeated 100 times and results were averaged.

##### Feature Ranking: XGB—Permutation Importance

An optimal XGB model was trained on all cases using all input features using 5-fold cross-validation and the area under the receiver operating curve (AUC) was determined. Then, the analyzed feature was shuffled, and the AUC was re-assessed. The importance of the analyzed feature was inferred from the difference in AUC between the two measurements. This was repeated 100 times and results were averaged.

##### Feature Ranking: XGB—Recursive Feature Elimination (RFE)

The XGB model was trained on all cases and features, and feature importance was assessed using the built-in XGB importance metric. Then, through iterations, the feature with the least importance was excluded, followed by re-training of the model on the remaining features [30]. By recursively eliminating each feature a time, a rank is given to each feature. This was repeated 100 times and results were averaged.

##### Feature Ranking: Average (AVG)

Ranking by averaging the results returned by the above four methods during the given iteration.

For each of the above methods, hyperparameters of the grid search are provided in Appendix A: Model hyperparameters for feature ranking.

##### Defining and Ranking Feature Groups

Taking into account the practical considerations related to an online risk calculator, we defined a minimum of four and a maximum of seven input features to be selected for the final model. Of note, attempts at using more than seven input features lead to marginal or no improvement in model metrics. Using the top four to seven features in each ranking, a total of 16 feature combinations were selected for assessment. These potential feature combinations were ranked by creating and evaluating machine learning models trained on them. Similarly to the approach used for ranking features, we aimed to apply multiple classifier types. RF, XGB, and MLP models were trained using 5-fold cross-validation (hyperparameters are provided in Appendix A: Model hyperparameters for feature group ranking. For each of the 16 feature groups, an “AUC score” was calculated by averaging of the top-performing 10 percent of the models trained on the given group (averaging was crucial to compensate for potential outlier values). Then, feature groups were ranked on this “AUC score”. The top-ranking feature group was selected both for the 1-month and for the 1-year endpoint.

#### 2.5.5. Model Selection

We sought to develop the optimal ML model for two classification tasks: (a) VT recurrence vs. no VT recurrence observed during the first month, and (b) VT recurrence vs. no VT recurrence observed during the first year following ablation. A flowchart of the model selection process is presented in Figure 2.

RF, XGB, MLP models were trained for both endpoints using the respective selected features. Each classifier type was tested with all combinations of the above-described scaling (standard scaling, min-max scaling, and no scaling) and oversampling (ROS, SMOTE, ADASYN) methods. A Bayesian grid search was performed for each classifier type, aiming to find the optimal hyperparameters through 100 iterations, using 10-fold cross-validation (i.e., 10 distinct 90–10% train-test splits). Hyperparameter spaces for the grid search are shown in Appendix A: Final model selection using Bayesian grid search. The best combination of pre-processing methods and hyperparameters was determined for each classifier type. Then, the top-performing models from each classifier type were compared based on the 10-fold cross-validation average AUC, and the best-performing approach was selected. Finally, we selected the fold of cross-validations (i.e., 10–90% train-test split) yielding the highest AUC as the final model. For both final models, the optimal threshold for classification had to be determined. Out of every possible threshold, the one with the highest Youden’s index was selected in order to maximize both sensitivity and specificity. In our final risk calculator (and in the SHAP analysis detailed below), we use this threshold to classify cases into “positive” vs. “negative” prediction with regard to the endpoint.

#### 2.5.6. Model Evaluation

Model performance was characterized with a range of metrics, in every case averaged across 10 cross-validations. Additionally to AUC, we specified commonly used metrics [31] of classification performance. Sensitivity (the proportion of true positives among actual positives), specificity (the proportion of true negatives among actual negatives), accuracy (the proportion of correct predictions to all predictions), positive likelihood ratio (the odds of obtaining a positive result in individuals meeting the endpoint compared to those who do not), and negative likelihood ratio (the odds of obtaining a negative result in individuals not meeting the endpoint compared to those who do) were calculated for each model.

Furthermore, we used Kaplan–Meyer analysis to compare VT-recurrence free survival of patients classified as positive vs. negative for (a) 1-month recurrence or (b) 1-year VT recurrence by the respective models. To prevent any potential data leakage, we determined the predicted outcome for each case by utilizing cross-validation, ensuring that the case in question was part of the testing set. To confirm that these sub-cohorts indeed show distinct VT-free survival, we used a Log-Rank test where *p* < 0.05 was considered statistically significant.

Shapley Additive Explanations (SHAP) analysis was used to calculate the Shapley values [32] of the features, which describe the relationship between each input feature and the model output. The relative importance of each feature’s contribution to the final output was assessed using the permutation importance method [29], and a radar chart was constructed to present this information.

#### 2.5.7. Comparison with the I-VT Score

We compared the performance of our 1-year VT recurrence prediction model to the I-VT score, a complex risk stratification system aimed to predict outcomes following VT ablation. We assessed each case in our database using the decision tree presented in Figure 4 of the original publication of Vergara et al. [15]. This model is aimed specifically to predict 1-year VT recurrence following ablation, using pre-procedural and procedural features. Only those cases where all required parameters were available were included (238/337, 70%).

We defined the features from our dataset corresponding to those used as input features for the decision tree in the following manner:Age, LVEF, CRT/ICD device: as recorded in our database.Previous ablation: “No” in all cases, since our cohort consisted of first ablations.Clinical VT inducible at the end of the ablation: “Yes”, if clinical VT was inducible before ablation and it was not successfully eliminated.Non-clinical VT inducible at the end of ablation: “Yes”, if non-clinical VT was inducible before ablation and it was not successfully eliminated.No VT inducible at the end of ablation: as recorded in our database.Regarding the findings of the final programmed extrastimulation, the input option “Not tested” in the I-VT score refers to cases where no programmed extrastimulation (PES) was performed. Since final PES was part of our protocol, this branch of the decision tree was not used in our cohort.

Then, we determined the results of the decision tree for each case in our cohort. The tree classified each case into a group; for each group, the average probability of 1-year VT recurrence was provided. Afterwards, an ROC curve was constructed using these predictions.

Finally, we used the DeLong unpaired test (R statistical software, R Foundation for Statistical Computing, Vienna, Austria—pROC package [33]), to assess for statistical difference between the ROC curve derived from the I-VT decision tree and the average AUC of our 1-year model.

## 3. Results

### 3.1. Patient Population

Out of the 337 patients included in our study, VT recurrence was observed in 117 (35%) cases in a median of 29 (4–111) days following the index procedure. Within one year of the index ablation, 68 (20%) patients died. The median age was 69 (60.2–74.8) years old, and most patients were male (295 (88%)). The most prevalent comorbidities were coronary artery disease (CAD) and chronic heart failure, present in 283 (84%) and 272 (82%) patients, respectively. Hypertension was present in 252 (75%) patients, while 113 (34%) patients had diabetes. Severely impaired functional status (New York Heart Association [NYHA] Class III or IV) was found in 101 (37%) patients. In addition, 61 patients (18%) had a history of SCD. Furthermore, 249 (74%) patients had a previously implanted ICD device for either primary or secondary prevention.

The majority of heart failure patients received optimal medical therapy according to the current guidelines. Beta-blockers (BB) were used in 233 (70%) cases, while amiodarone was given to 302 (91%) patients.

Baseline parameters are shown in Table 1. Notably, patients with 1-year VT recurrence had larger left ventricular diameters (LVEDD *p* < 0.001, LVESD *p* < 0.001), worse left and right ventricular function (LVEF, *p* = 0.008; TAPSE, *p* = 0.028), more severe mitral regurgitation (*p* = 0.007), worse NYHA functional status (*p* = 0.027), and more often had a previously implanted CRT (*p* < 0.001) or ICD (*p* = 0.042) device. They more frequently presented with ICD shocks (*p* = 0.001), or electrical storm (*p* = 0.02), and on average they showed multiple inducible VT morphologies (*p* < 0.001) during the procedure. Patients with recurrence within the first month had similar characteristics; interestingly, unsuccessful elimination of the clinical VT during the procedure was more common in these patients.

### 3.2. Machine Learning Analysis

The steps of the entire feature selection process, along with the features excluded during each step, are detailed in Appendix A: Feature selection overview. The results of the 100 iterations using each feature ranking method are shown in Appendix A: Feature ranking results. Feature rankings obtained through the four different methods are presented in Table 2. Feature group ranking can be found in Appendix A: Feature group ranking results.

The optimal input feature combination was defined for the 1-month and the 1-year recurrence endpoint. Common input features between the 1-year and 1-month models were hemodynamic instability and ICD shock. Features specific to 1-month recurrence were the presentation with incessant VT, LVEF, TAPSE, and the inducibility of the clinical VT at the end of the procedure. Features specific to 1-year recurrence were the number of inducible VT morphologies, LVESD, and mitral regurgitation. The highest-performing 30 models from each combination of pre-processing, oversampling, and model type are presented in Appendix A: Final model selection results. The highest achievable AUC for each classifier type, along with the pre-processing approach used, is presented in Table 3.

For both endpoints, the best performance was demonstrated by a random forest model. For the 1-year endpoint, no pre-processing or oversampling was used; for the 1-month endpoint, standard scaling was used in conjunction with SMOTE. The average AUC of the 10 cross-validations is presented for each model. Evaluating the classification performance on the test populations, an AUC of 0.73 was determined for 1-month recurrence, while the 1-year recurrence model had an AUC of 0.71 (Figure 3). In contrast, the average performance on the test population was 0.76 and 0.75, respectively. Additional model metrics are provided in Table 4).

We conducted Kaplan–Meyer analysis to examine recurrence-free survival. In Figure 4a, we compared patients with VT recurrence within 1 month to those without; in Figure 4b, we did the same for patients with 1-year VT recurrence. In both cases, the two groups show significantly different VT-free survival (Log-rank *p* < 0.001 in both cases).

We also compared our algorithm to an established prediction system for 1-year recurrence, the I-VT score [15], on our test population. According to the AUC metric, our model provided better predictions for 1-year recurrence post ablation (AUC 0.52 vs. 0.71, (DeLong Test *p* = 0.0002) (Figure 5).

We used the permutation importance approach to assess the relative importance of each of the input features (Figure 6). Apparently, the most important feature predicting recurrence in the first month is hemodynamic instability, while the number of VT morphologies contributes the most to 1-year outcome. ICD shock appears to have the smallest contribution to the model prediction out of the features assessed.

Finally, we used SHAP analysis to explain how the individual features affect the final model output (Figure 7). From the figure, it can be inferred that feature values and their effect on model output show a clear correlation. Additionally, the effect of a given feature is consistent across cases, i.e., a higher value (or the presence) of a feature always shifts model prediction in the same direction. Looking at the first figure, we can conclude that poor left and right ventricular function shifts model prediction towards 1-month VT recurrence. Similarly, presentation with incessant or hemodynamically unstable VT, or an adequate ICD shock increases the probability of early recurrence being predicted. Meanwhile, the elimination of the clinical VT reduces the predicted probability of this endpoint. Analysis of the 1-year VT recurrence classifier showed that a more dilated left ventricle, more severe mitral regurgitation, and more inducible VT morphologies increase the predicted risk of the endpoint. Similarly, presentation with hemodynamic instability and ICD shocks appear to predict 1-year VT recurrence.

## 4. Discussion

Our single-center registry study aimed to implement a machine-learning pipeline for the accurate prediction of VT recurrence after catheter ablation in patients with structural heart disease, with 1-year follow-up.

Through feature selection using a range of machine learning approaches, we defined the optimal set of features to predict 1-month and 1-year VT recurrence. Given the moderate sample size, we were able to implement a range of resource-intensive machine learning approaches for feature selection. Interestingly, the optimal input features for the 1-month random forest model were derived from a ranking based on permutation importances calculated from an MLP model during the feature ranking process, emphasizing the importance of using diverse algorithms for feature selection.

Using the selected features, we trained various machine learning models, out of which the random forest approach proved the most efficient. The resulting models were assessed using 10-fold cross-validation. Using Kaplan–Meyer survival analysis on the test population, we concluded that the VT-free survival differs significantly between the two groups defined using our model. Furthermore, we identified the most important features in predicting VT-recurrence. The permutation importance metrics (Figure 6, radar charts) and the SHAP analysis (Figure 7) represent two distinct approaches to interpreting model prediction. Permutation importance accurately reflects the contribution of a feature to overall model performance (i.e., the correct predictions). Conversely, Shapley values represent the contribution of a feature to model predictions in general, and additionally reflect the relationship (positive/negative) between the feature value and the resulting prediction. In our data, we see good congruence of the two approaches (e.g., for the 1-year model both figures show that the number of VT morphologies is the most important, while ICD shock is the least important). From the statistical study, it stands clear that every one of the selected input features shows a statistically significant difference respective to the endpoint assessed.

The prediction of VT recurrence after catheter ablation has emerged to be a field of interest in recent years [34], and recurrent VT has become a well-established predictor of mortality [11]. In a recent meta-analysis of seminal randomized controlled trials in the field, 1-year VT-free survival following ablation was 66.3%, which dropped to 55.5% when assessed at 5 years [35]. Several factors have been associated with VT recurrence, including lower LVEF [36], multiple inducible VT morphologies [37], impaired NYHA functional status [37], presentation with an electrical storm [37,38], and non-ischemic VT [38]. The largest cohort study to date was conducted by Tung et al., where non-ischemic cardiomyopathy, advanced NYHA status, previous ICD implantation or cardiac resynchronization therapy (CRT), lower left ventricular ejection fraction (LVEF), electrical storm, and ICD shocks were the most important predictors of VT recurrence [11]. A more recent prospective study assessed 281 consecutive patients with non-ischemic cardiomyopathy undergoing VT ablation. The authors identified ICD shocks, basal anteroseptal VT origin, and procedural failure as predictors of VT recurrence [12]. In contrast to the previous study, a recent, smaller analysis of 50 cases included a majority of ischemic heart disease patients [13]. This study revealed that along with atrial fibrillation and advance heart failure, female sex appears to pose a higher risk of post-ablation VT recurrence. This, however, was not confirmed in one of our studies specifically aimed at sex-related differences in outcomes after VT ablation [14]. Additionally, the timing of the procedure appears to be of great importance. Although early ablation appears to have certain benefits in terms of reducing VT burden [39], randomized trials designed to assess prophylactic ablation are mostly inconclusive [40,41].

The parameters selected during our feature selection process greatly overlap with known predictors of VT recurrence. Specifically, LVEF, clinical VT inducibility after ablation, ICD shocks, and the number of inducible VT morphologies have all been previously associated with lower success rates [11,12,37,42,43,44].

Most previous studies with the exception of one [37] mainly focused on 1-year recurrence. In our cohort, we found that a significant number (50%) of recurrence events occurred within the first month after ablation, emphasizing the importance of our two separate endpoints. Few studies exist that specifically assess early VT recurrence. Notably, in a work with 370 patients, early recurrence, defined as recurrent arrhythmia within a week, was predicted using advanced heart failure status, DCM, VT storm, and a greater number of induced VTs [37]. Importantly, mortality was significantly higher in the early recurrence group, underscoring the clinical relevance of VT recurrence early after the procedure.

The aforementioned studies included large numbers of patients; however, they are limited by the constraints of conventional survival analysis. It is known that non-linear relationships, multi-collinearity, and complex feature interactions can only be analyzed by more advanced statistical approaches. Indeed, it has been proven in a range of studies that machine learning provides a way to uncover more intricate correlations between factors and can ultimately lead to more accurate predictions [45,46]. Vergara et al. previously created a prediction system for post-ablation VT recurrence. LVEF, device therapy, non-ischemic DCM, and previous ablation were the key factors associated with recurrence, along with VT inducibility during the procedure. Of note, in their sample population, non-ischemic DCM and previous ablation were significantly more common, compared to our cohort. Their decision-tree-based approach is easily interpretable and provides reasonably good predictions. However, decision trees are prone to overfitting, meaning that they perform well on the sample data, but can be less accurate on new data. This, along with the different baseline parameters, might explain why our prediction algorithm outperforms the I-VT score on our data.

We expect our prediction system to be a valuable tool in clinical practice. All variables required to make a prediction are easily acquired through history-taking and routine examinations.

## 5. Limitations

Conclusions and applicability of the present study is to some degree limited by the sample size. The analyzed registry includes all eligible cases of VT ablation performed in a large-volume tertiary center. Nonetheless, the analysis of a larger cohort would understandably lead to higher predictive power and more favorable model metrics. Furthermore, the external validation of the presented machine learning models could further strengthen our conclusions.

Due to the small sample size, initial train–test splitting was not feasible. Attempts aimed at initial train–test splitting yielded unacceptably high variance of model metrics, depending on the random seed used to create the split. Given this limitation, we applied measures to prevent overfitting and data leakage; K-fold cross-validation was applied during feature selection and also during the selection of the final classifiers.

An additional limitation of our analysis is related to the high mortality in the assessed population. Classification methods, like the ones presented here, cannot handle censoring, i.e., patients who deceased during the follow-up period. Consequently, in 36 of the included cases, no VT recurrence was observed, yet the patient was deceased for another reason within one year of the procedure. This may have impacted the accuracy of the models trained on our data.

## 6. Conclusions

In our study, we used state-of-the-art machine learning methods to develop a reliable risk stratification system, accurately predicting post-ablation VT recurrence in patients with structural heart disease. By entering easily obtainable pre-procedural and procedural features, the user receives a prediction of estimated 1-month and 1-year mortality.

## Figures and Tables

**Figure 1 bioengineering-10-01386-f001:**
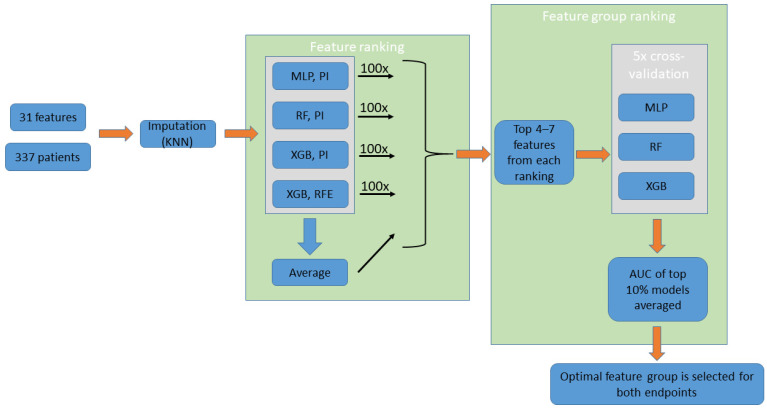
Feature selection (performed separately for the two endpoints). The process uses the imputed data of 17 features of the entire patient population. Four feature ranking methods are performed (100 iterations averaged, not shown on the figure), while a fifth ranking order “Average” is obtained from averaging the results from the four methods. Groups created using the top 4–7 features of each ranking are assessed using ML models. The groups are then ranked based on the average AUC of the top 10% models trained on them. Finally, the top-ranking feature group is selected. KNN: K-nearest neighbors; MLP: multilayer perceptron; RF: random forest; XGB: extreme gradient boosting; PI: permutation importance; RFE: recursive feature elimination.

**Figure 2 bioengineering-10-01386-f002:**
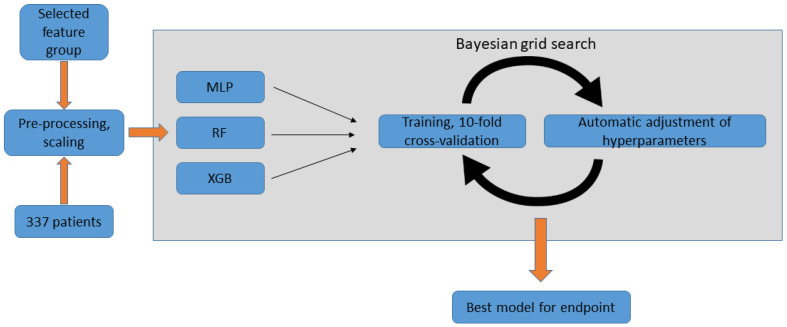
Model selection using Bayesian grid search, performed for each endpoint. Pre-processed data of the selected features is used by the Bayesian grid search algorithm for each of the three classifier types. In each cycle, the model is tested using 10-fold cross-validation and the hyperparameters are automatically adjusted. After a pre-defined number of iterations, the process returns the final model. MLP: multilayer perceptron; RF: random forest; XGB: extreme gradient boosting.

**Figure 3 bioengineering-10-01386-f003:**
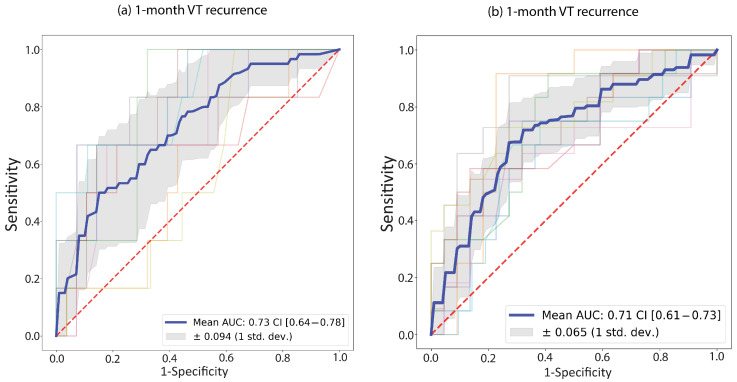
Receiver Operating Curves (ROC) for the two random forest models; (**a**) 1-month recurrence model; (**b**) 1-year recurrence model. ROC curves are plotted for each fold in the 10-fold cross-validation (faded colored lines); the blue line represents their average. The grey shaded area shows the standard deviation of the ten ROC curves. The legend shows the area under the average ROC (“Mean AUC”) with the 95% confidence interval (“CI”).

**Figure 4 bioengineering-10-01386-f004:**
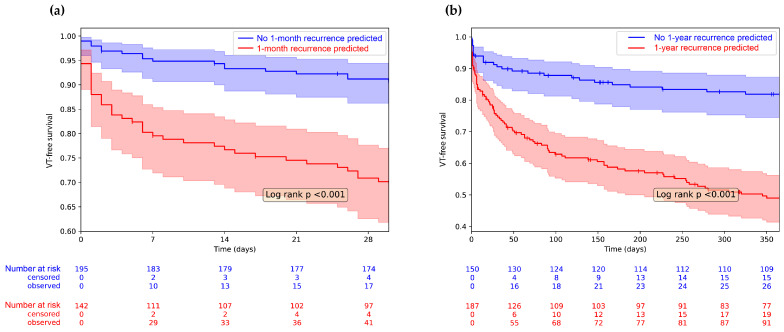
Kaplan–Meyer analysis showing 1-year VT recurrence based on the model prediction. Censored cases are symbolized using vertical lines on the survival curves. (**a**) We divided the population based on the prediction of the 1-month VT recurrence model and the VT-free survival of the two groups is plotted with 1-month follow-up; (**b**) The entire population was divided based on the prediction of the 1-year VT recurrence model and the VT-free survival of the two groups is plotted with 1-year follow-up. Shaded areas show 95% confidence intervals. VT-free survival of the two groups is compared with the Log-Rank test; the inset shows the *p* value.

**Figure 5 bioengineering-10-01386-f005:**
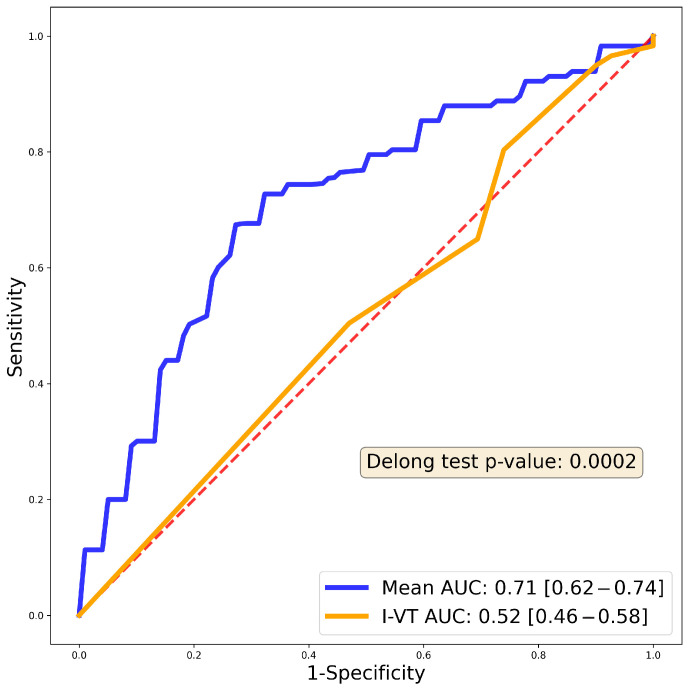
Receiver Operating Curve (ROC) of the previously published I-VT score on our patient population compared to our 1-year predictive model. Blue curve shows the 1-year random forest AUC, while the orange curve shows the I-VT score AUC (0.52). The red dashed line is the reference line. DeLong test shows significantly better performance in the case of our predictive model (*p* value indicated in the inset).

**Figure 6 bioengineering-10-01386-f006:**
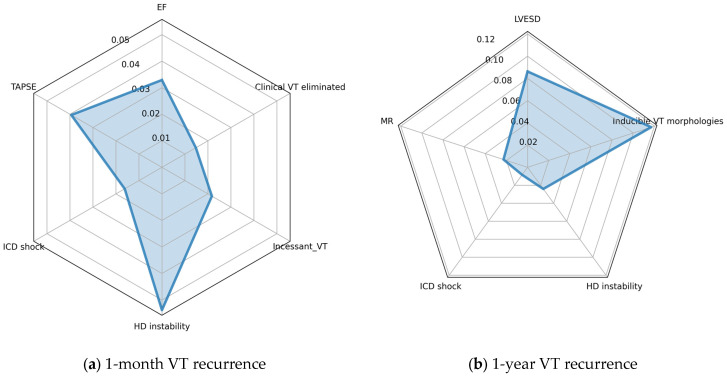
Radar charts showing the relative importance of input features, as determined using the permutation approach; (**a**) 1-year recurrence model; (**b**) 1-month recurrence model. Values denote the reduction in mean AUC after shuffling the values of the given feature. EF: left ventricular ejection fraction; VT: ventricular tachycardia; TAPSE: tricuspid annular plane systolic excursion; ICD: implanted cardioverter defibrillator; LVESD: left ventricular end systolic diameter; HD: hemodynamic; MR: mitral regurgitation.

**Figure 7 bioengineering-10-01386-f007:**
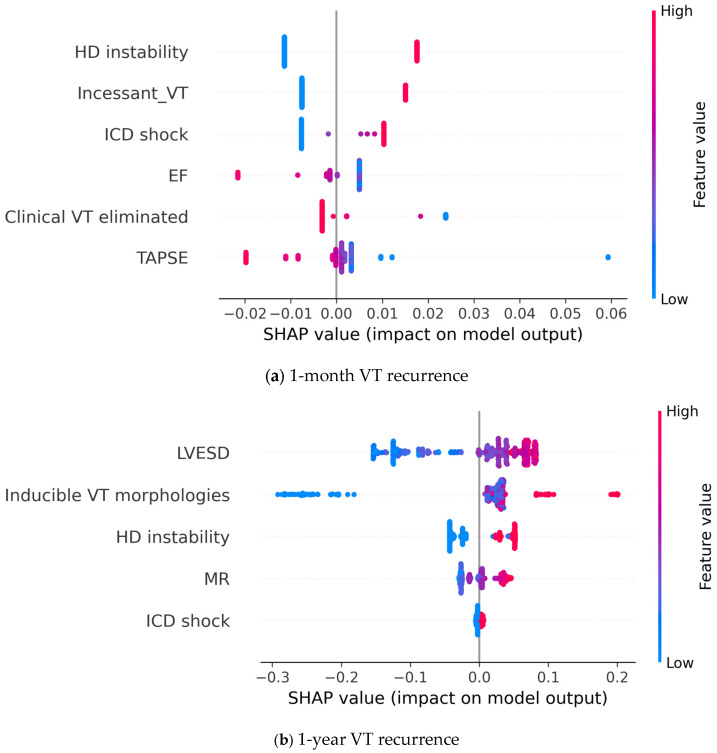
SHAP analysis showing the contribution of individual input features to the model prediction output; (**a**) 1-year recurrence model; (**b**) 1-month recurrence model. Each point represents a patient case, where the color denotes the value of a given feature and the SHAP value shows the contribution of that given feature to the model output. EF: left ventricular ejection fraction; VT: ventricular tachycardia; TAPSE: tricuspid annular plane systolic excursion; ICD: implanted cardioverter defibrillator; LVESD: left ventricular end systolic diameter; HD: hemodynamic; MR: mitral regurgitation.

**Table 1 bioengineering-10-01386-t001:** Baseline characteristics: For categorical variables, count (percent) are presented, for scalar variables median [interquartile range] is presented. *p* < 0.05 was considered statistically significant.

	Data Available N (%)	All Patients N = 337 (100%)	1-Month VT Recurrence			1-Year VT Recurrence		
			1-Month VT Recurrence N = 60 (18%)	No 1-Month VT Recurrence N = 277 (82%)	*p*	1-Year VT Recurrence N = 117 (35%)	No 1-Year VT Recurrence N = 220 (65%)	*p*
Age	337 (100%)	68.7 (60.2–74.8)	68.2 (61.6–76.1)	68.7 (60.1–74.8)	0.619	67.5 (61.1–74.7)	68.9 (59.3–75.0)	0.934
Male	337 (100%)	295 (88%)	51 (85%)	244 (88%)	0.659	102 (87%)	192 (88%)	1.000
Atrial fibrillation	337 (100%)	109 (32%)	25 (42%)	84 (30%)	0.126	45 (38%)	64 (29%)	0.109
Hypertension	337 (100%)	252 (75%)	43 (72%)	209 (76%)	0.622	83 (71%)	169 (77%)	0.261
Diabetes	337 (100%)	113 (34%)	18 (30%)	95 (34%)	0.613	43 (37%)	70 (32%)	0.445
COPD	337 (100%)	42 (12%)	8 (13%)	34 (12%)	1.000	13 (11%)	29 (13%)	0.697
CAD	337 (100%)	283 (84%)	48 (80%)	235 (85%)	0.426	95 (81%)	188 (86%)	0.339
ICD	337 (100%)	249 (74%)	47 (78%)	202 (73%)	0.508	95 (81%)	154 (70%)	0.042 *
CRT	337 (100%)	80 (24%)	20 (33%)	60 (22%)	0.081	42 (36%)	38 (17%)	<0.001 *
HF	331 (98%)	272 (82%)	53 (88%)	219 (81%)	0.234	100 (85%)	172 (80%)	0.314
NYHA	306 (91%)	2 (1–3)	2 (1–3)	2 (1–3)	0.120	2 (1–3)	2 (1–3)	0.027^*^
SCD	337 (100%)	61 (18%)	12 (20%)	49 (18%)	0.823	19 (16%)	42 (19%)	0.605
EF	305 (91%)	34 (27–42)	30 (25–35)	35 (27.2–43)	0.012 *	33 (25–38)	35 (28–44.2)	0.008 *
LVESD	283 (84%)	50 (42–57)	54 (47–60)	48.5 (41–56)	0.011 *	53 (47–59)	47 (41–54)	<0.001 *
TAPSE	264 (78%)	19 (15–23)	17 (14–20)	19 (16–23)	0.019 *	18 (14–20.2)	19 (16–23)	0.028 *
E wave DT	262 (78%)	162 (133–213)	150 (123–193)	165 (137–220)	0.083	150 (127–200)	170 (140–220)	0.008 *
MR	299 (89%)	2 (1–2)	2 (1–3)	2 (1–2)	0.036 *	2 (1–3)	2 (1–2)	0.007 *
TR III-IV	278 (82%)	34 (12%)	8 (15%)	26 (12%)	0.592	13 (13%)	21 (12%)	0.881
Amiodarone	331 (98%)	233 (70%)	47 (80%)	186 (68%)	0.118	88 (76%)	145 (67%)	0.140
Beta blocker	331 (98%)	302 (91%)	53 (90%)	249 (92%)	0.867	106 (91%)	196 (91%)	1.000
ICD shock	333 (99%)	140 (42%)	33 (57%)	107 (39%)	0.018 *	60 (52%)	80 (37%)	0.009 *
HD instability	337 (100%)	132 (39%)	36 (60%)	96 (35%)	0.001 *	58 (50%)	74 (34%)	0.007 *
Incessant VT	337 (100%)	114 (34%)	30 (50%)	84 (30%)	0.006 *	52 (44%)	62 (28%)	0.004 *
Electrical storm	336 (100%)	135 (40%)	35 (58%)	100 (36%)	0.003 *	57 (49%)	78 (36%)	0.027 *
Inducible VT morphologies	334 (99%)	1 (1–2)	1 (1–2)	1 (1–2)	0.166	1 (1–2)	1 (1–2)	0.001 *
Clinical VT inducible	333 (99%)	261 (78%)	49 (83%)	212 (77%)	0.431	104 (90%)	157 (72%)	<0.001 *
Non-clinical VT(s) inducible	332 (99%)	106 (32%)	21 (36%)	85 (31%)	0.609	38 (33%)	68 (31%)	0.909
Clinical VT cycle length	268 (80%)	400 (340–460)	400 (342–450)	400 (340–460)	0.828	400 (354–458)	386 (333–458)	0.114
Clinical VT eliminated	321 (95%)	284 (88%)	44 (79%)	240 (91%)	0.020 *	100 (88%)	184 (88%)	1.000
All VTs eliminated	322 (96%)	255 (79%)	40 (70%)	215 (81%)	0.095	91 (80%)	164 (79%)	0.950
Major complications	334 (99%)	32 (10%)	10 (17%)	22 (8%)	0.061	15 (13%)	17 (8%)	0.186

COPD: chronic obstructive pulmonary disease; SCD: sudden cardiac death; CAD: coronary artery disease; ICD: implantable cardioverter defibrillator; CRT: cardiac resynchronization therapy; HF: heart failure; NYHA: New York Heart Association functional status; LVEF: left ventricular ejection fraction; LVESD: left ventricular end systolic diameter; TAPSE: tricuspid annular plane excursion; MR: mitral regurgitation; HD: hemodynamic. The asterix (*) denotes statistical significance.

**Table 2 bioengineering-10-01386-t002:** Feature ranking for the 1-month (left sub-table) and 1-year (right sub-table) endpoints. For each ranking method (column name), the average rank across 100 iterations is shown. An additional ranking based on the average of ranks across the four methods is shown in the AVG column. Both sub-tables are sorted by the AVG column.

1-Month VT Recurrence					
	AVG	XGB, RFE	MLP, PI	RF, PI	XGB, PI
HD instability	1	0	0	0	6
LVEF	2	1	4	2	1
TAPSE	3	7	1	3	4
Age	4	3	13	4	0
Clinical VT eliminated	5	2	3	5	12
LVESD	6	4	12	1	5
ICD shock	7	9	2	7	9
Clinical VT cycle length	8	6	14	6	2
E wave DT	9	5	10	11	3
Electrical storm	10	8	6	9	8
MR	11	10	8	8	10
Incessant VT	12	11	5	13	11
Inducible VT morphologies	13	16	7	10	13
NYHA	14	13	15	15	7
All VTs eliminated	15	12	11	14	15
Atrial fibrillation	16	15	9	16	14
TR III-IV	17	14	16	12	16
1-year VT recurrence					
	AVG	XGB, RFE	MLP, PI	RF, PI	XGB, PI
LVESD	1	1	4	0	0
Inducible VT morphologies	2	0	0	1	7
MR	3	4	1	2	6
ICD shock	4	6	2	3	9
Age	5	2	13	5	1
E wave DT	6	5	6	7	3
LVEF	7	3	12	8	2
HD instability	8	8	3	4	10
TAPSE	9	7	7	9	4
Incessant VT	10	9	5	11	11
Clinical VT cycle length	11	10	16	6	5
NYHA	12	11	8	10	8
All VTs eliminated	13	13	10	12	13
TR III-IV	14	12	14	13	12
Electrical storm	15	14	9	16	14
Atrial fibrillation	16	15	11	14	15
Clinical VT eliminated	17	16	15	15	16

HD: hemodynamic; LVEF: left ventricular ejection fraction; VT: ventricular tachycardia; LVESD: left ventricular end systolic diameter; DT: deceleration time; TAPSE: tricuspid annular plane systolic excursion; ICD: implantable cardioverter defibrillator; MR: mitral regurgitation; TR: tricuspid regurgitation; XGB, RFE; Extreme gradient boosting sing recursive feature elimination; MLP, PI: Multilayer perceptron combined with permutation importances; RF, PI: random forest combined with permutation importances; XGB, PI: Extreme gradient boosting combined with permutation importances; AVG: average.

**Table 3 bioengineering-10-01386-t003:** Best combinations of scaling, pre-processing, and hyperparameters, shown from each classifier type. The upper table shows the 1-month VT recurrence endpoint, while the lower table shows the 1-year VT recurrence endpoint. RF: random forest; MLP: multilayer perceptron; XGB: extreme gradient boosting; SS: standard scaling; SMOTE: Synthetic Minority Oversampling Technique; VT: ventricular tachycardia; AUC: area under the curve.

1-month VT recurrence	Model	Pre-Processing	Oversampling	Hyperparameters	Mean AUC (Test)
	RF	Not used	Not used	RF__class_weight: None; RF__criterion: log_loss; RF__max_depth: 1; RF__max_features: 1; RF__min_samples_leaf: 2; RF__n_estimators: 300; RF__random_state: 0	0.730
	MLP	SS	SMOTE	NN__activation: logistic; NN__alpha: 0.00225; NN__hidden_layer_sizes: 3; NN__max_iter: 161; NN__random_state: 0; NN__solver: adam	0.729
	XGB	Not used	Not used	XGB__alpha: 7.99; XGB__colsample_bytree: 1.0; XGB__gamma: 0.1; XGB__max_depth: 3; XGB__min_child_weight: 0.0; XGB__random_state: 0; XGB__scale_pos_weight: 2.0; XGB__subsample: 0.5	0.708
1-year VT recurrence	Model	Pre-processing	Oversampling	Hyperparameters	mean AUC (test)
	RF	SS	SMOTE	RF__class_weight: balanced_subsample; RF__criterion: gini; RF__max_depth: 2; RF__max_features: 7; RF__min_samples_leaf: 2; RF__n_estimators: 320; RF__random_state: 100	0.713
	XGB	Not used	Not used	XGB__colsample_bytree: 1.0; XGB__gamma: 9.79; XGB__max_depth: 3; XGB__min_child_weight: 0.0; XGB__random_state: 0; XGB__scale_pos_weight: 3.14; XGB__subsample: 0.897	0.711
	MLP	SS	SMOTE	NN__activation: logistic; NN__alpha: 0.000192; NN__hidden_layer_sizes: 50; NN__max_iter: 50; NN__random_state: 0; NN__solver: adam	0.709

**Table 4 bioengineering-10-01386-t004:** Metrics for both the 1-year and 1-month model. Values shown are the average of the 10 cross-validations. SS: standard scaling; SMOTE: Synthetic Minority Oversampling Technique; AUC: area under the receiver operating curve; LR+: positive likelihood ratio; LR−: negative likelihood ratio.

	Model Type	Scaling	Pre-Processing	AUC (Test)	AUC (Train)	Accuracy	Sensitivity	Specificity	LR+	LR−
1-month	Random forest	Not used	Not used	0.730	0.758	0.68	0.63	0.7	3.2	0.53
1-year	Random forest	SS	SMOTE	0.713	0.751	0.71	0.61	0.77	2.8	0.5

## Data Availability

The data presented in this study are available on request from the corresponding author.

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
