# Peer review of "Machine-Learning-Based Prediction of 1-Year Arrhythmia Recurrence after Ventricular Tachycardia Ablation in Patients with Structural Heart Disease"

_bioengineering, 2023, doi:10.3390/bioengineering10121386_

Round 1

Reviewer 1 Report

Comments and Suggestions for Authors

Great work, with sound statistics acceptable machine learning criteria. What was my concern was

1. almost 340 ablations for 17 years, this is too low.

2. And in conclusion maybe suggestion that this prediction could be gain if there was enough statistical power to validate it on a larger number. one center observational is rather Hypothesis generating than strong conclusive trial

Author Response

We highly appreciate the valuable comments and made an effort to revise the manuscript in accordance. Please see attachment.

Reviewer 2 Report

Comments and Suggestions for Authors

In this study, the authors employed machine learning to predict ventricular tachycardia (VT) recurrence one month and one year post VT ablation. The dataset included medical history, laboratory tests, echocardiography, and surgical data from 337 patients who underwent VT ablation. Despite the comprehensive workflow of the article, several questions arise:

 1. On line 131, the authors used four feature ranking methods, each repeated 100 times. Could you provide details on how the model parameters were configured for each of the 100 experiments for each ranking method and what the experimental results were?

2. On line 133, the authors selected the top 4 to 7 features, resulting in 16 sets. Why was the number of features chosen to be between 4 and 7, and is there an exploration of accuracy for feature sets with more than 7 features?

 3. On line 145, the authors mentioned using different oversampling methods for preprocessing. What is the purpose of employing various oversampling methods?

 4. On line 161, the authors stated that 68 patients died within one year. How were these 68 deceased patients handled in the data, and what classification results were assigned to them?

 5. Supplementary Tables 5 and 6 present data where most parameters and experimental results are identical. Could you provide information on the actual models used and the corresponding experimental outcomes?

 6. The authors employed four feature ranking methods and three machine learning models during feature selection but ultimately chose random forests for the final model. If only the built-in random forest feature ranking algorithm was used during feature selection, followed by training and testing with the random forest model alone, would this approach yield better results? Please include additional experiments to address this question.

Comments on the Quality of English Language

no more

Author Response

We highly appreciate the Reviewer’s time and rigor to elaborate insightful comments and made an effort to revise the manuscript in accordance. Please find our detailed response in the attachment.

Reviewer 3 Report

Comments and Suggestions for Authors

This single-center clinical trial to evaluate short-term (1 month) and long-term (1 year) outcome in patients with ventricular tachycardia after ablation used detailed clinical data collected over a nearly 17-year period (2005-2022). Important findings from machine learning algorithms are certainly important to improve the survival rate of such patients who do not have a good prognosis according to the current data, which identifies a one-year mortality rate of 20%. The data are worthy to be published, however, there are important issues in the manuscript content (text, tables and figures) which should be improved before taking a positive decision for the publication. The detailed list of major revision remarks below cover almost all sections of the manuscript: Abstract, Introduction, Methods, Results and Discussion.

Major revision remarks:

1.       Ln 25, “Results: 1-year VT recurrence was observed in 117 (35%) cases. Our 1-month prediction ….”,  Ln 28 “Input features for the 1-year prediction….” -> The number of patients with 1-month VT recurrence is missing. In general, the logical presentation of the results is confusing as a mix of 1-month and 1-year prediction features. It is better to split the results of the two tasks into different parts, especially if different ML models are trained for these tasks.

2.       Abstract: Add one more sentence with information about the type of the ML classifier.  

3.       Abstract: The proportion of patients split for training and test of the algorithms is not disclosed. This issue is of major importance in any ML task to ensure that training results are not reported but ML models are tested on independent patients. The ML algorithms are prone to overtraining and the training cases must be sufficiently large and exceeds the number of input features. Furthermore, define the global number of input features and the methods for feature importance ranking. The ML classifier itself does not provide this information. The authors should also check the main text so that these important issues are sufficiently described wherever necessary in section Methods.

4.       Introduction: The introduction is very short and includes an insufficient number of 11 references, which are mostly before 2017. Authors should supplement their section Introduction with new research in the field. The focus may be on the design of clinical trials, the identified important characteristics and techniques for evaluating them – similar to the area of interest in this study.

5.       Ln 73-74: Study approval is an important matter that requires identification of the Ethics Committee institution, date and decision identification number. Also identify if the study was registered in a public register, such as ClinicalTrials or similar.

6.       Sub-sections in 2. Materials and Methods must be numbered and properly formatted, including “Patient population”, “Ablation procedure”, “Baseline statistics”, “Machine Learning analysis”.

7.       Missing description of the features in section “2. Materials and Methods”. A dedicated section is required to systematically describe all 96 features prior to their mention in Ln 121.

8.       Ln 121-122: “From the 96 features 17 were selected based on previous literature and clinical relevance.” -> The following statement is confusing. First, it is necessary to identify all sources of this important information for the methodological design. Such manual selection of 17 out of 96 features has no sense for the ML classifier. It is not setup into the aims. Furthermore, revealing 96 features is meaningless and it is questionable why it is necessary to measure them all, given that only 17 are preliminary (in a pre-study) identified important. This manual selection is not identified in Figure 1.

9.       Ln 120: “>30% missing data were excluded” -> This important condition is not identified in Figure 1. Furthermore, ensure that excluded features are not among the preliminary chosen 17.

10.    Ln 122: “Missing values imputed using the K-nearest neighbors method.” -> It is necessary to give more information about this approach. Why and how used! K-nearest neighbors can give different results based on the number of output classes and input set. Furthermore, the use of the same ML classifier for missing measurements filling can surely lead to biased results.

11.    Section “Baseline statistics” should cover all necessary information about “Statistical study” (not limited to baseline statistics). The methodological design should identify the groups of the patient population used in the statistical comparisons as well as for the training and testing of the classifier. If this is a binary classification task, then two independent groups must be identified. For example, Group 1.1 (patients without VT recurrence within 1 month) vs. Group 2.1 (patients with 1 month VT recurrence); Group 1.2 (patients without VT recurrence within 1 year) vs. Group 2.2 (patients with 1 year VT recurrence).

12.    Section “Machine Learning analysis” is quite incomprehensive and insufficient. Just referring “RF, XGB, and MLP methods” is not an issue of publication. The methodological design and hyperparemeter settings of each algorithm should be systematically explained. The idea is that each algorithm can be reproduced in other studies. The whole section should be written from the scratch.

13.    Ln 126: The statement “Feature importance was assessed and ranked using four different algorithms” is in fact not associated with any of the three classifiers “RF, XGB, and MLP” in Ln 128 or 6 other classifiers “RF, XGB, MLP, Logistic regression, K-Nearest Neighbors, and Support Vector Clasifier” in Ln 141. The reader is confused as to exactly what was done in the methods and whether the experimental setup was correct. The classifiers themselves do not output feature importance but other algorithms for evaluation of their learned hidden rules are necessary. It is strongly required that classifiers and algorithms for feature importance evaluation are described separately in details.

14.    Figure 1: The block diagram does not correspond well to the text in section “Machine Learning analysis”. This figure must be considerably changed, increasing its informative value. Moreover, use consistent information in the figure and in the text such as : “10-fold cross-validation” vs. 5-fold cross-validation in Ln 136.

15.    Figure 1: “Model with best AUC is selected and trained” -> What exactly was done? The text is confusing -Retraining an already trained model? Retraining other training data or using the test data for training? How many models are finally trained?

16.    Section “Machine Learning analysis”: It is good to provide formula for the feature importance selection. Furthermore, the source and mathematical knowledge of each algorithm should be identified with respective reference. The programming environment (hardware and software platform) as well as the use of special software libraries should be identified.

17.    Ln 159, Table 1 header line -> recurrence of what?

18.    Ln 162-164: Check the most prevalent comorbidities, the first appears to be CAD in 84% of the population.

19.    Table 1: Reported median [95% CI] but the statistical design (Ln 114) and baseline characteristics (Ln 159-160) report medians [interquartile ranges]. Apply consistent statistical analyses in the study, which should be in line with the statistical design.

20.    Table 1: In the header line, the authors need to report the number of patients in each group because the results cannot be explained. For example: 51 (85%) vs. 244 (88%).

21.    Table 1: Identify statistical differences with some mark (e.g. "*") and do not rely on bolded values, as bolding may not always be recognized and depends on formatting and the system font.

22.    Ln 190-195. Supplements: In general the provided supplement in unreadable and useless. These are excel electronic tables that contain unexplained and unformatted tables mainly in compressed machine learning output report. Most of them are incomprehensive and unreadable for the general public. Authors must carefully revise and select ONLY the most valuable supplementary information, if needed, and include in an all-in-one pdf file, where each table has a clear caption; each header row has a clear explanation and units of reported data; each cell has comprehensive numerical/categorical data. For example, Ln. 190: “Feature ranking was performed using four different methods (Supplementary Table 1).” -> This information is of crucial importance for the results and must be included in the main text. The data in the table are incomprehensive and need careful explanation.

23.    Explain more about the selected features: Are the same features selected by the different ML algorithms and do they match the baseline statistics with significant features in Table 1?

24.    Ln 200-201: Do not duplicate information already communicated in methods, otherwise confusing.

25.    Ln 205-206: “For both endpoints, the best performance was demonstrated by a random forest model.” -> Since different ML methods are defined in Methods, the reader expects to see a numerical ranking of the results of different classifiers and then objectively evaluate how good the random forest is compared to others. This information is omitted and presents questions as to whether other models have been implemented and properly used/reported.

26.    Ln 206-207: „For the 1-year endpoint, no preprocessing or oversampling was used; for the 1-month endpoint, standard scaling was used, but no oversampling.” -> More information is needed in Methods as it does not include comprehensive methodological details about “preprocessing”, “oversampling” and “scaling” with sufficient details of their importance and implementation.

27.    Figure 2, caption: “The legend shows the mean AUC with the 95% confidence intervals”, however, the visible legend denotes Mean AUC and +/-1 std. dev. Which one is correct? Report consistent statistical metrics according to the definition in the methodological section “Statistical design”. Furthermore, “the average is shown in blue” is in conflict with mean AUC: Is this the same?

28.    Table 2 reports metrics, which are NOT explained in methods. They cannot be interpreted in the context of this study.   

29.    Ln 218-222: The paragraph should be expanded to include more information on the methods used, their significance and data analysis. Information should be appropriately divided into Methods and Results as soon as Results cannot cover numerical values that are not explained in Method how to derive them.

30.    Figure 3: The label “Log rank p<0.001” is not explained in Methods, and it cannot be interpreted in the context of this study. Its presence in the figure is quite confusing.

31.    Figure 4: More information in the text is needed to explain how the “orange” curve is obtained. The term “I-VT score on our patient” is not interpretable.

32.    Figure 6 and explanation in Ln 239-244: Carefully revise the description of the feature importance seen in Figure 6 because the text does not comprehensively reveal important observations seen in the figure. Authors should carefully explain and emphasize the importance of this figure.

33.    Section Discussion misses comparative study with other researchers in the field: Numerical values of performance reported in this study vs. other teams.

Author Response

We greatly appreciate the time and rigor of the Reviewer in providing genuinly insightful comments. We made an effort to revise the manuscript in accordance. Please, find our specific comments in the attachment.

Round 2

Reviewer 3 Report

Comments and Suggestions for Authors

After the first review round, the authors have substantially improved the quality of their manuscript. Further minor revision remarks are disclosed:

1.       Ln 55-56, “Multiple factors have been identified that contribute to the success of the procedure [11–14].” -> This generalized single paragraph has no sense. More details should be provided on all factors identified as important.

2.       Ln 60: Reference [19] is mentioned but the number is not sequential to the previous reference [14]. Correct such inconsistency if present elsewhere in the text.

3.       The aims of the study must be defined in a separate paragraph. Do not mix the paragraph with the literature review.

4.       Ln 162-163: The  K-Nearest  Neighbors (KNN) imputer” and Ln 185 -> The term “imputer” is not correct. KNN is known as a clustering method or a classifier.

5.       Ln 179: “From the 29 features 17 were manually selected” -> The entire statement is confusing and can't be interpreted because up to this point in the article, readers have not been informed of any kind of extracted features. The entire section must be moved after the feature description (ensure that all 29 features were disclosed) and before Figure 1.

6.       All Tables and Figures must be placed just after their first reference in the text (not several pages later). The section “3.3. Figures, Tables” is meaningless and does not represent a proper structure of a scientific paper.

7.       Ln 303: “decision tree presented in Figure 4” -> Noted inconsistency: a decision tree is NOT depicted in Figure 4!!! Furthermore, make sure that figures presenting results are not referenced in section Methods.

8.       Figures 6 and 7 present results of the study and must be first referenced in section Results. Additional comments can be further provided in section Discussion.

Author Response

We appreciate the insightful comments and made an effort to modify the manuscript accordingly. Please find detailed answers in the attachment.
